# LOST AT THE BEGINNING OF REASONING

## ABSTRACT

Recent advancements in large language models (LLMs) have significantly advanced complex reasoning capabilities, particularly through extended chain-of-thought (CoT) reasoning that incorporates mechanisms such as backtracking, self-reflection, and self-correction. Despite these developments, the self-correction abilities of LLMs during long CoT reasoning remain underexplored. And recent findings on overthinking suggest that such models often engage in unnecessarily redundant reasoning. In this work, we empirically show that the first reasoning step exerts a disproportionately large influence on the final prediction—errors introduced at this stage can substantially degrade subsequent reasoning quality. This phenomenon is consistently observed across various state-of-the-art open- and closed-source reasoning models. Leveraging this insight, we propose an efficient sampling strategy that leverages a reward model to identify and retain high-quality first reasoning steps while discarding suboptimal ones, achieving up to a 70% reduction in inference cost without sacrificing accuracy.

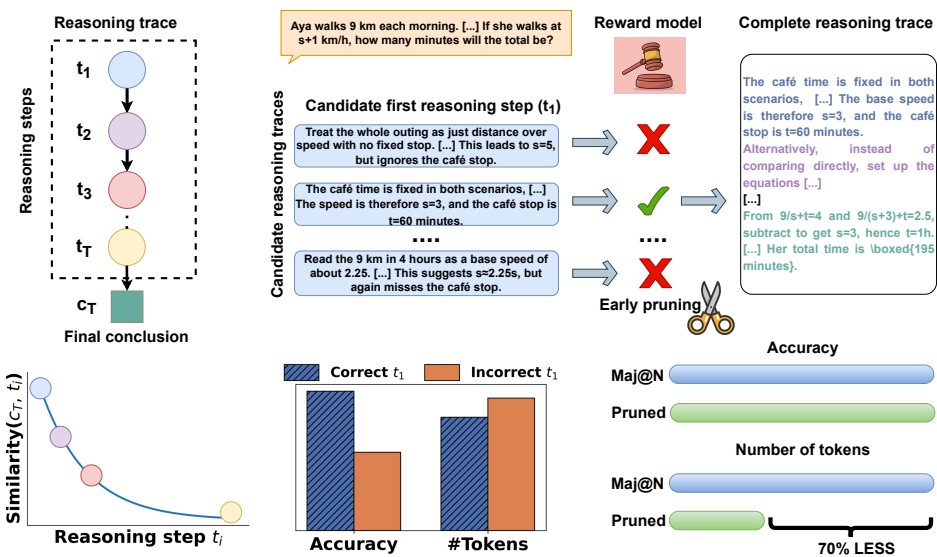

Figure 1: Overview of our observation and efficient sampling. The first reasoning step $t_1$ heavily shapes the entire reasoning trajectory: a strong first step typically yields accurate solutions with fewer tokens (bottom left). Building on this observation, we propose to generate multiple candidate first steps, evaluate them with a reward model, and discard weaker candidates early (top right). This method maintains accuracy while substantially reducing token consumption by 70% (bottom right).

## 1 INTRODUCTION

Large language models (LLMs) have demonstrated remarkable performance across a variety of reasoning tasks, ranging from mathematical problem solving to multi-hop question answering (Hestness et al., 2017; Kaplan et al., 2020; Hoffmann et al., 2022). More recently, the advent of reasoning-oriented LLMs capable of performing long chain-of-thought (long-CoT) reasoning at test time has

led to substantial advancements in these domains (Brown et al., 2020; Hurst et al., 2024; Anthropic, 2025; Team et al., 2024; Guo et al., 2025; Yang et al., 2025a; Wen et al., 2025; He et al., 2025). A widely held hypothesis attributes this progress to the models' ability to backtrack, self-reflect, and self-correct, effectively leveraging contextual feedback to iteratively refine their responses.

However, recent studies suggest that long-CoT reasoning can also introduce inefficiencies. Models often "overthink" by producing unnecessarily extended reasoning processes to solve problems (Chiang & Lee, 2024; Zhang et al., 2024a; Wang et al., 2025b; Liao et al., 2025b;a). This observation raises questions about the model's capacity for backtracking, self-reflection, and self-correction. It suggests that LLMs may lack awareness of the information they have already processed, leading to redundant or inefficient reasoning. Moreover, Liu et al. (2024a) demonstrate that LLMs are prone to the "lost-in-the-middle" phenomenon, wherein information located in the middle of a long context is often overlooked. While their analysis is conducted in the context of information retrieval, it remains an open question whether similar positional biases affect long CoT as well.

In this work, we introduce a novel and previously underexplored perspective on long-CoT reasoning: many reasoning failures in long-CoT LLMs stem not from errors made mid-chain, but rather from flaws at the beginning of reasoning. Our experiments demonstrate that the first reasoning step has the most significant influence on the final prediction. When this first step is incorrect, the model is considerably more likely to arrive at an incorrect final answer (40% accuracy drop), highlighting the limited self-correction capabilities of current long-CoT LLMs. Notably, this phenomenon is consistently observed from five open- and closed-source long-CoT LLM families.

Motivated by this insight, we propose an efficient early pruning algorithm that exploits the predictive power of the first reasoning step. Specifically, by evaluating the quality of the first step, we identify and discard less promising reasoning traces early, continuing generation only for the more promising ones. This approach significantly reduces inference cost. Across five open-sourced long-CoT LLM families and five challenging mathematical, scientific reasoning and programming benchmarks, our method maintains accuracy while reducing inference budget by up to 70%. Our results show that the first step is not just the beginning of reasoning, but a key factor that influences both accuracy and efficiency, making it an important focus for future reasoning models.

**Contributions.** Our main contributions are as follows: (1) To the best of our knowledge, we firstly empirically establish a strong positive correlation between the first reasoning step and the final prediction across various open- and closed-sourced long-CoT LLM families (§3); (2) Inspired by this observation, we propose an efficient early pruning algorithm that halts generation for less promising initial steps, thereby improving inference efficiency while maintaining the accuracy (§4); (3) Both observation and proposed efficient sampling method are extensively validated on various long-CoT LLMs across different reasoning tasks, with necessary control experiments to disentangle the confounding factors.

## 2 RELATED WORK

**Lost in the middle.** Liu et al. (2024a) introduced the "lost in the middle" effect, demonstrating that LLMs tend to overlook information in the middle of long contexts, performing better when relevant content appears at the beginning or end. This positional bias is evident across tasks like arithmetic reasoning (Shen et al., 2023; Liao & Monz, 2024), multiple-choice QA (Zheng et al., 2024; Pezeshkpour & Hruschka, 2023), text evaluation (Wang et al., 2024; Shi et al., 2024), passage ranking (Zhang et al., 2024b), and instructional prompt positioning (Liu et al., 2024b; Chen et al., 2024b). Additionally, studies have documented primacy and recency biases, where models disproportionately allocate attention to the first or final tokens, independent of their semantic relevance (Xiao et al., 2024; Qin et al., 2023; Barbero et al., 2025). While previous studies have primarily examined positional biases in external context, we investigate whether analogous biases emerge in internal reasoning trajectories of long chain-of-thought models. Different from attention-level analyses that focus on how the first input token shapes representations, our work shows that the first generated reasoning step greatly influences subsequent reasoning and final outcomes.

**Efficient test-time reasoning.** Test-time scaling methods aim to improve the accuracy–compute trade-off by adapting sampling and aggregation. One line of work increases self-consistency efficiency by reducing sample counts (Li et al., 2024; Wan et al., 2025; Aggarwal et al., 2023; Xue et al., 2023),

while another shortens chain-of-thought depth via fine-tuning or inference-only optimizations (Chen et al., 2024a; Luo et al., 2025; Hou et al., 2025; Fu et al., 2025a; Yang et al., 2025b). These methods, however, still rely on generating full reasoning traces. DeepConf (Fu et al., 2025b) instead uses local confidence to filter low-quality traces and terminate generation early. Our method takes a different focus: we assess the quality of the initial reasoning step, which strongly shapes subsequent reasoning, and prune weak starts before long traces unfold.

## 3 LOST AT THE BEGINNING OF REASONING

Motivated by the finding of Liu et al. (2024a), which demonstrates that query-relevant information is more impactful when positioned at either the beginning or end of an LLM's context window, we first investigate whether a similar positional effect exists in long-CoT reasoning (§3.1). Our analysis reveals that the first reasoning step has great impact to the final conclusion. To validate this observation, we further perform two ablation studies, confirming the critical role of the first step in determining the model's final prediction (§3.2 and §3.3).

**Notation.** Let $p$ represent the input prompt, consisting of both a system instruction and a user query. A reasoning model $\mathcal{M}$ produces a sequence of CoT reasoning steps $t = [t_1, t_2, ..., t_T]$, followed by a final conclusion $c_T$, such that the complete model output is given by $t \oplus c_T = \mathcal{M}(p)$, where $\oplus$ means concatenation. In models such as DeepSeek-R1 (Guo et al., 2025) and Qwen3 (Team, 2025), the input-output format adheres to the following:

$$p <\text{think}> t_1, t_2, \ldots, t_T </\text{think}> c_T$$

The final prediction $q_T$ is then derived by applying an extraction function $g$ to the conclusion, i.e., $q_T = g(c_T)$, where $g$ may, for example, extract values enclosed within \boxed{}.

The conclusion $c_T$ can be interpreted as a summary of the essential reasoning steps leading to the final prediction. This raises an interesting question: *Is there a positional bias in how reasoning steps contribute to the conclusion?* In other words, do certain steps have a disproportionately greater influence on $c_T$ than others?

### 3.1 SIMILARITY BETWEEN REASONING STEPS AND THE FINAL CONCLUSION

To understand how different reasoning steps contribute to the final conclusion, we measure the semantic similarity between each reasoning step $\{t_i\}_{i=1}^T$ and the final conclusion $c_T$.

To assess how intermediate reasoning contributes to the final outcome, we measure the semantic similarity between each reasoning step $\{t_i\}_{i=1}^T$ and the final conclusion $c_T$. This analysis reveals whether the reasoning process gradually aligns with the correct answer or diverges along the way.

**Experimental setup.** We evaluate 60 questions from AIME24 and AIME25 (MAA Committees, 2025) using DeepSeek-R1-Distill-Qwen-7B (abbreviated as DS-R1-Qwen-7B in the remainder of this paper) (Guo et al., 2025), Qwen3-8B (Yang et al., 2025a), Claude-3.7-Sonnet with thinking (Anthropic, 2025), GPT-OSS-20B (Agarwal et al., 2025), and Magistral-Small (Rastogi et al., 2025).[1] Generations are produced with `temperature=1.0`, `top_p=0.9`, `min_p=0.05`, and `max_tokens=32768`; for Claude-3.7-Sonnet, only `max_tokens` is set. All subsequent experiments adopt this hyperparameter configuration.

**Segmentation of reasoning steps.** We define a reasoning step as a complete logical leap or self-contained unit (Xiong et al., 2025), and segment reasoning traces with GPT-5.[2] To complement this setup, we also employ heuristic segmentation based on reasoning switch keywords (Wang et al., 2025a), with details provided in Appendix C.

**Similarity computation.** We compute semantic similarity between each step $t_i$ and the conclusion $c_T$ by taking the cosine similarity of their embeddings obtained from all-MiniLM-L6-v2 (Reimers & Gurevych, 2019; Wang et al., 2020). To avoid inflated similarity from problem restatement, we use GPT-5 mini to remove question-overlap text at the beginning of traces. As a robustness check,

---

[1]For reproducibility, the exact model identifiers are provided in Appendix D.

[2]By default, we use GPT-5 mini for step segmentation; for GPT-OSS-20B, we instead use GPT-5, as the mini variant is incompatible.

we also report results with SPLADE similarity (Formal et al., 2021) in Appendix E, confirming that our findings are not specific to dense embeddings. Since traces vary in length, similarity curves are interpolated to a fixed number of steps (either the average or maximum length) for visualization.

This setup allows us to capture how reasoning trajectories semantically converge toward—or deviate from—the final answer across different models.

**Result.** Figure 2 shows that the first reasoning step exhibits the highest similarity to the final conclusion, after which similarity drops sharply. Beyond the initial few steps, similarity stabilizes at a lower level, with only minor fluctuations across the remainder of the reasoning process. These results suggest that the first step $t_1$ is most closely aligned with the final conclusion and likely sets the overall direction of the reasoning. Subsequent steps appear to introduce exploratory or redundant content that deviates from the final answer. Additional results using SPLADE similarities (Figure E.2) show the same trend, confirming that this pattern is robust across similarity metrics. Taken together, these findings show that the first reasoning step plays a disproportionately important role in shaping the final conclusion.

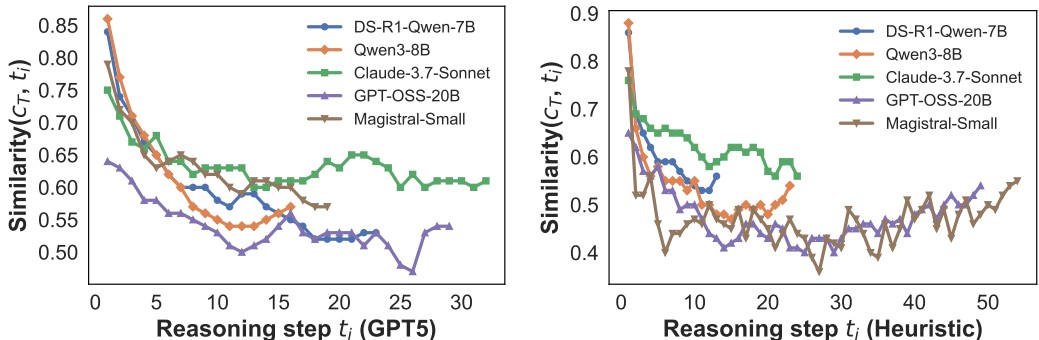

Figure 2: Cosine similarity between the embeddings of the $i$-th reasoning step $t_i$ and the final conclusion $c_T$, using the average number of reasoning steps for interpolation. The reasoning steps are segmented either by GPT-5 (left) or by heuristic rules (right). See Figure E.1 for results based on the maximum number of reasoning steps used for interpolation.

*Given the strong alignment between early reasoning steps—particularly the first—and the final conclusion, we hypothesize that the first step may significantly influence whether the reasoning model can arrive at a correct prediction.*

## 3.2 CORRELATION BETWEEN THE FIRST REASONING STEP AND THE FINAL PREDICTION

Given that the first reasoning step closely resembles the final conclusion, we investigate whether the essential reasoning required for the final prediction is already encapsulated in the first step. To this end, we analyze the prediction when conditioned solely on the first reasoning step. Specifically, we compute $c_1 = \mathcal{M}(p\text{<think>}t_1\text{</think>})$, and derive the corresponding prediction $q_1 = g(c_1)$, which we compare against the ground truth $a$. Based on this comparison, we categorize each first reasoning step as either `first correct` (if $q_1 = a$) or `first incorrect` (otherwise).

**Experimental setup.** To better analyze the correlation, we sample 64 CoT traces per question using the same datasets in §3.1. We exclude questions for which all 64 CoT traces result in either correct or incorrect predictions, as these are considered either too easy or too difficult, respectively, yielding 38 questions for DS-R1-Qwen-7B and 37 for Qwen3-8B. For each remaining question and its corresponding first reasoning step $t_1$, we obtain the initial prediction $q_1$ as previously described. While GPT-5 provides more reliable segmentation, it is costly and difficult to reproduce. We therefore adopt the heuristic segmentation method in all subsequent experiments, which is shown to have comparable results with GPT5 segmentation in §3.1. To better visualize the final outcomes, we categorize the questions into three groups based on the pass@1 accuracy of the final prediction $q_T$,[3] corresponding to the intervals (0, 33%], (33%, 67%], and (67%, 100%). A higher pass@1 indicates a

---

[3]That is, the average accuracy across 64 CoT traces.

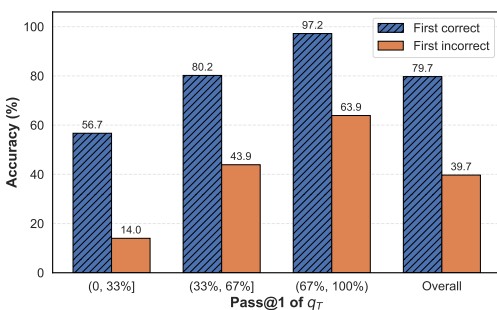 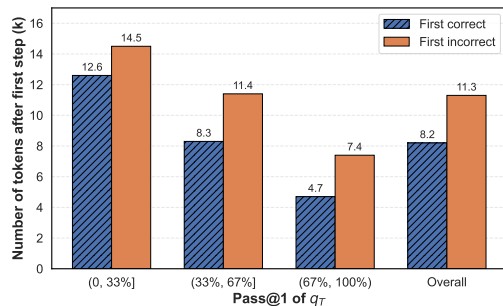

Figure 3: Accuracy and number of tokens on DS-R1-Qwen-7B. **Left:** The relationship between the accuracy of the final prediction ($q_T$) and the correctness of the prediction solely based on the first reasoning step ($q_1$) across different difficulty levels. If $q_1$ is incorrect, $q_T$ is more likely incorrect. **Right:** The number of tokens used for the final prediction after the first reasoning step $t_1$, i.e., the number of tokens used for $[t_2, t_3, ..., t_T]$. Although $q_1$ is correct, the model still consumes a large amount of tokens for the following reasoning steps–overthinking.

simpler question. This grouping allows us to assess whether our observations hold consistently across varying levels of question difficulty.

**Result.** As shown in Figure 3 (Left), the commonly assumed self-correction capability of reasoning models appears to be overstated. When the first reasoning step is incorrect, the model's final prediction is also likely to be incorrect. On average, final prediction accuracy drops by 40% when the first reasoning step is incorrect, with the most substantial decrease (43%) occurring for difficult questions (0–33% range) and a notable decline (33%) even for easier questions (67–100% range). In addition, we also compute the Pearson correlation between the correctness of the first prediction $p_1$ and the final prediction $p_T$ over all questions. The coefficient $r = 0.60$ and p-value $p = 0.0$ denote a moderately strong positive correlation. All these results underscore the pivotal role of the first reasoning step in steering the model toward a correct final answer, particularly in more complex instances where recovery from early mistakes is more challenging. Extending this analysis to DeepSeek and Qwen models of different sizes yields consistent trends: final accuracy remains substantially higher when the first step is correct, and the accuracy gap persists as model scale increases (Figure F.2).

Figure 3 (Right) further illustrates the model's tendency toward overthinking. Even when the first reasoning step is sufficient to arrive at the correct answer, the model continues to generate a substantial number of additional reasoning tokens—the same scale in length to those generated following an incorrect first step. Both findings are also observed for Qwen3-8B in Figure F.1, reinforcing this pattern across models.

### 3.3 MINOR PERTURBATION TO THE CORRECT FIRST STEP LEADS TO SIGNIFICANT LOSS

Building on our findings in §3.2, which demonstrate a positive correlation between the model's first and final predictions, we further investigate the significance of the first reasoning step by introducing minor perturbations. Specifically, we slightly alter an initially correct reasoning step and provide it as input to the model to assess whether it can recover from such errors.

**Experimental setup.** Unlike §3.2, where we analyze the correctness of the first reasoning step $t_1$, here we treat the final correct conclusion $c_T$—which satisfies $q_T = g(c_T) = a$—as the new first reasoning step, denoted $t'_1$. This choice ensures that the step contains all necessary reasoning for arriving at the correct answer, which cannot be guaranteed for $t_1$. As illustrated in Figure 3 (Left), an initially correct reasoning step can still lead to an incorrect final prediction. To construct $t'_1$, we apply the following perturbations to $c_T$ (see Appendix I for an example): (1) we remove not only the explicit answer formatting (e.g., \boxed{a}) but also any surrounding sentences that may directly disclose or repeat the final answer; (2) the resulting text from (1) is treated as the correct version of $t'_1$ (serving as our baseline); (3) we generate an incorrect version by replacing the correct answer $a$ in the remaining reasoning with $a \pm 1$ or $a \pm 10$.[4]

---

[4]The answer of AIME question is integer in the range of $[0, 999]$.

These perturbations are minimal, as they preserve the core reasoning structure while only altering the final prediction in the incorrect variant. We then combine the prompt $p$ with the modified first reasoning step $t_1'$ and input it to the model as $\mathcal{M}(p\text{<think>}t_1'\text{Alternatively})$ to assess subsequent reasoning behavior.

**Result.** As shown in Table 1, we make two key observations: (1) Smaller models rarely reach 100% accuracy even when the first reasoning step is correct,

Table 1: Perturbation experiments. Reported accuracy (%) with correct vs. incorrect first step. Even minor perturbations cause significant drops.

| Model | Correct (%) | Incorrect (%) |
|---|---|---|
| DS-R1-Qwen-1.5B | 95.4 | 64.4 |
| DS-R1-Qwen-7B | 94.8 | 28.5 |
| DS-R1-Qwen-32B | 100.0 | 85.8 |
| Qwen3-1.7B | 96.0 | 46.6 |
| Qwen3-8B | 71.4 | 37.0 |
| Qwen3-30B-A3B | 100.0 | 74.7 |
| Qwen3-235B-A22B | 100.0 | 78.7 |

suggesting that they may revise or deviate from their initial reasoning. In contrast, larger models (e.g., DS-R1-32B) consistently achieve 100% accuracy given a correct first step, indicating greater stability. (2) There is a substantial drop in accuracy when the first reasoning step is incorrect, highlighting that even minor errors early in the reasoning process can significantly affect the final prediction. These findings further indicate that the LLM's ability to self-correct has been overestimated.

In this section, we observe that the reasoning model is particularly vulnerable at the initial stage of the reasoning process; an error in the first step can propagate and substantially degrade the final prediction. *Can we develop a method to identify and retain more promising first reasoning steps while discarding suboptimal ones to enhance the overall generation efficiency?*

# 4 EARLY PRUNING WITH HINT FROM FIRST STEP

In this section, we propose an efficient and straightforward sampling method to identify a promising first reasoning step. By doing so, we can terminate the generation process early when a suboptimal first step is detected, thereby reducing unnecessary computational overhead.

## 4.1 PROBLEM DEFINITION

In contrast to the notation introduced in §3, we incorporate a random seed $\epsilon$ to introduce stochasticity into the sampling process. Specifically, a sampled trace is computed as $t \oplus c_T = \mathcal{M}(p, \epsilon)$. By varying the random seed $\epsilon^n$, we obtain diverse generations, denoted as $t^n \oplus c_T^n = \mathcal{M}(p, \epsilon^n)$, where $t^n = [t_1^n, t_2^n, \ldots, t_T^n]$.[5]

A widely adopted technique for reasoning tasks is majority voting or self-consistency generation (Wang et al., 2022). To promote exploration of the reasoning space, models are typically sampled at high temperatures, resulting in diverse outputs. Majority voting then serves to aggregate these outputs by selecting the most frequent final prediction. Formally, majority voting over $K$ samples is defined as:

$$q_{\text{maj}@K} = \text{mode}(\{q_T^n\}_{n=1}^K) \quad \text{where} \quad q_T^n = g(c_T^n)$$

However, for models generating long CoT traces, majority voting becomes computationally expensive, as it requires sampling $N$ complete reasoning paths independently. In this section, we propose a more efficient approach that samples only $M$ full traces, where $M < N$, while maintaining comparable majority voting performance to the one with $N$ samplings.

## 4.2 METHODOLOGY

In §3, we demonstrated a strong positive correlation between the first reasoning step and the final prediction. This observation motivates a method that identifies the top $M$ most promising first reasoning steps out of a total of $N$, and continues generation only for these selected $M$ candidates, while discarding the remaining $(N - M)$.

Let a reasoning model generate $N$ candidate first reasoning step $\{t_1^1, t_1^2, \ldots, t_1^N\}$ from a prompt $p$ with different random seeds $\{\epsilon^n\}_{n=1}^N$. Each $t_1^n$ is the first reasoning step of a full reasoning trajectory.

---

[5]In prior experiments, we sampled 64 CoT traces per question using 64 distinct random seeds $\{\epsilon^n\}_{n=1}^{64}$.

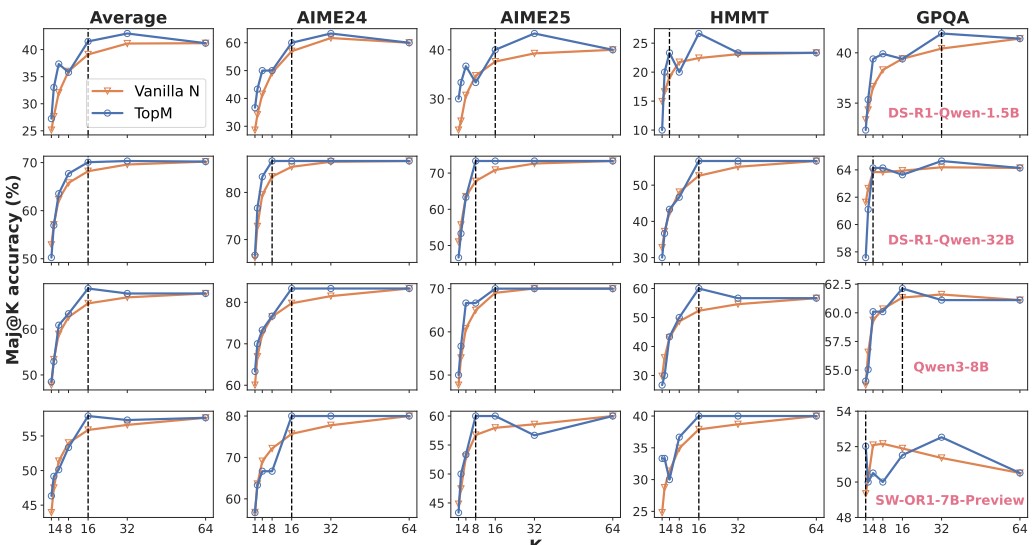

Figure 4: Majority voting accuracy with different number of samplings for four LLMs. The vertical dashed line denotes the smallest $M$ whose accuracy is equal to or larger than the accuracy of $N = 64$.

We define a scoring function $r : t_1^n \to \mathbb{R}$ that estimates the promise of a first step, e.g., rating from a reward model. We then select the top $M$ first steps based on their scores:

$$\mathcal{R}_{\text{top}} = \text{TopM}(\{r(t_1^n)\}_{n=1}^N)$$

Only the selected $M$ first steps $\{t_1^n \mid n \in \mathcal{R}_{\text{top}}\}$ are used for further multi-step generation. The remaining $(N - M)$ are discarded. Since the first step typically requires only a small number of tokens to generate, this selection approach efficiently reduces computation by avoiding full sampling for the less promising $(N - M)$ candidates.

### 4.3 EXPERIMENTS

**Setup.** We evaluate five families of reasoning models—DS-R1-Qwen (Guo et al., 2025), Qwen3 (Yang et al., 2025a), Skywork-OR1 (SW-OR1) (He et al., 2025), Magistral (Rastogi et al., 2025) and GPT-OSS (Agarwal et al., 2025)—on five challenging reasoning benchmarks spanning mathematics, science and programming: AIME24, AIME25 (MAA Committees, 2025), HMMT Feb 2025 (Balunović et al., 2025), the GPQA Diamond set (Rein et al., 2024) and LiveCodeBench[6] (Jain et al., 2024), consisting of 30, 30, 30, 198, and 175 problems, respectively. For decoding, we adopt each model's recommended temperature and top_p, with a maximum generation length of 32K tokens.

We consider values of $N$ and $M$ from the set $\{1, 2, 4, 8, 16, 32, 64\}$. For early pruning, we fix $N = 64$ and select the top $M$ most promising first-step candidates using reward scores predicted by a process reward model (PRM), Qwen2.5-Math-PRM-7B (Zhang et al., 2025). When $M = 64$, the accuracy is exactly the same as the one for $N = 64$, since all candidates are chosen. For PRM, a "step" is defined as a segment of text ending with "\n\n", and each step receives an individual reward score. The trajectory $t_1^n$ contains multiple such steps, and we use the score of its final step as the overall reward, $r(t_1^n)$. Notably, using a PRM to score $t_1^n$ is really cheap, because $t_1^n$ is short, and its computation is similar to generate one token with input $p \oplus t_1^n$.

Unlike the definition used in §3, where $t_1^n$ terminates upon generating the keyword "Alternatively", we redefine $t_1^n$ in this section to have a fixed token length, $\text{len}(t_1^n)$. The previous definition made it difficult to control the computational budget for generating first steps, as the trigger phrase might not appear or could occur late in the sequence. By fixing the token length, we achieve precise control over the generation budget when sampling $N$ candidate first steps. By default, $\text{len}(t_1^n) = 512$.

---

[6]Latest release, containing problems from Jan–Apr 2025.

Table 2: Early pruning accuracy and efficiency. We select $M = 16$ first steps with the highest reward scores out of $N = 64$ candidate first steps. The number of tokens used for the 48 discarded first steps is also included for the early pruning method. Early pruning maintains the accuracy, even improves sometimes, while only requiring $< 30\%$ original inference budget.

| Model | Method | Maj@K accuracy (%) ↑ | | | | | Avg. #Tokens ↓ |
| | | AIME24 | AIME25 | HMMT | GPQA | Avg. | |
|---|---|---|---|---|---|---|---|
| DS-R1-Qwen-1.5B | $N = 64$ | **60.0** | **40.0** | 23.3 | **41.4** | 41.2 | ×1.00 |
| | $N = 16$ | 57.0 | 37.6 | 22.4 | 39.4 | 39.1 (-2.1) | ×0.25 |
| | $M = 16$ | **60.0** | **40.0** | **26.7** | 39.4 | **41.5** (+0.3) | ×0.28 |
| DS-R1-Qwen-32B | $N = 64$ | **86.7** | **73.3** | **56.7** | **64.1** | **70.2** | ×1.00 |
| | $N = 16$ | 85.4 | 70.9 | 52.6 | 63.9 | 68.2 (-2.0) | ×0.25 |
| | $M = 16$ | **86.7** | **73.3** | **56.7** | 63.6 | 70.1 (-0.1) | ×0.29 |
| Qwen3-8B | $N = 64$ | **83.3** | **70.0** | 56.7 | 61.1 | 67.8 | ×1.00 |
| | $N = 16$ | 79.8 | 69.0 | 52.3 | 61.3 | 65.6 (-2.2) | ×0.25 |
| | $M = 16$ | **83.3** | **70.0** | **60.0** | **62.1** | **68.9** (+1.1) | ×0.28 |
| SW-OR1-7B | $N = 64$ | **80.0** | **60.0** | **40.0** | 50.5 | 57.6 | ×1.00 |
| | $N = 16$ | 75.7 | 58.0 | 37.9 | 51.9 | 55.9 (-1.7) | ×0.25 |
| | $M = 16$ | **80.0** | **60.0** | **40.0** | **52.5** | **57.9** (+0.3) | ×0.29 |
| Magistral-Small | $N = 64$ | 86.7 | **83.3** | **76.7** | **70.2** | **79.2** | ×1.00 |
| | $N = 16$ | **87.1** | 82.6 | 71.1 | 69.2 | 77.5 (-1.7) | ×0.25 |
| | $M = 16$ | 86.7 | **83.3** | 73.3 | **70.2** | 78.4 (-0.8) | ×0.28 |
| GPT-OSS-20B | $N = 64$ | **86.7** | **83.3** | **80.0** | **73.2** | **80.8** | ×1.00 |
| | $N = 16$ | 85.3 | 81.7 | 73.3 | 72.7 | 78.3 (-2.5) | ×0.25 |
| | $M = 16$ | **86.7** | **83.3** | **80.0** | **73.2** | **80.8** (-0.0) | ×0.27 |

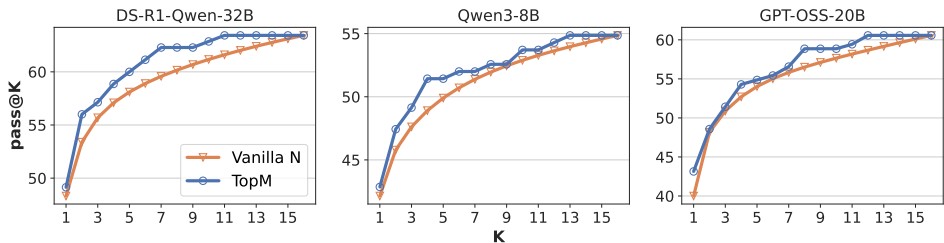

Figure 5: The pass rate on LiveCodeBench, where we set $N = 16$ for early pruning (TopM).

### 4.3.1 MAIN RESULTS

In Figure 4, we analyze the performance as $M$ varies. We find that selecting the top 16 first reasoning steps from 64 candidates and continuing generation from them achieves accuracy on par with, or even exceeding, conventional sampling with $N = 64$. This trend is consistent across diverse LLMs and benchmarks. Interestingly, for certain cases—such as HMMT on DS-R1-Qwen-1.5B, and AIME24, AIME25, and GPQA on DS-R1-Qwen-32B—using as few as $M \leq 8$ suffices to match the performance obtained with all 64 samples.

Table 2 reports the detailed accuracy and token consumption of different methods. When using $M = 16$, early pruning consistently matches the accuracy of $N = 64$ across a range of LLMs and benchmarks, while substantially outperforming $N = 16$ under a comparable token budget. Notably, for Qwen3-8B, early pruning even yields a 1.1% improvement in average accuracy. Importantly, these gains come at less than 30% of the inference cost of majority voting with $N = 64$, underscoring the strong efficiency advantage of our method.

For the code generation benchmark, LiveCodeBench, where majority voting is not applicable, we present the pass rate in Figure 5. Early pruning consistently surpasses standard sampling given the same number of complete sampling.

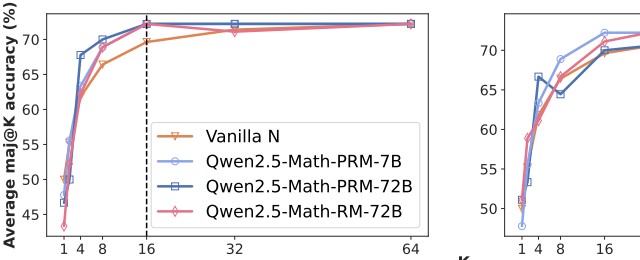

Figure 6: Average accuracy across AIME24, AIME25 and HMMT on DS-R1-Qwen-32B. **Left:** Comparison of reward signals derived from different reward models. The choice of reward model has minimal impact on overall performance. **Right:** Effect of varying the length of the first reasoning step. Using a very short first step, like 256 tokens, leads to suboptimal performance, likely because it provides insufficient reasoning context to effectively evaluate the quality of the step.

### 4.3.2 ABLATION STUDIES

Here we further validate our default settings.

**Choice of reward model.** In Figure 6 (Left), we evaluate two additional reward models: a larger PRM, Qwen2.5-Math-PRM-72B, and an outcome reward model, Qwen2.5-Math-RM-72B (Yang et al., 2024). The results indicate that the choice of reward model has minimal impact on performance. Notably, the smaller 7B PRM achieves comparable results, highlighting the efficiency of our approach.

**Length of first step.** In Figure 6 (Right), we examine the impact of varying the length of the first reasoning step. We observe that the shortest first step (i.e., $\text{len}(t_1^n) = 256$) leads to degraded performance. We hypothesize that shorter $t_1^n$ sequences lack sufficient reasoning content, making them less informative for reliable reward model evaluation. Nevertheless, setting $\text{len}(t_1^n) \geq 512$ tokens yields consistently better performance than vanilla sampling.

**Effect of first step split.** Table 3 examines how defining the first step influences early pruning performance. The heuristic approach follows the method described in §3. Overall, both definitions yield comparable results. Nevertheless, we recommend using the token-count–based definition, as it provides a clearer way to manage the token budget across candidate first steps. Moreover, trigger phrases that signal step boundaries may vary across LLMs.

Table 3: Average maj@K for early pruning, with first step defined by length or phrase.

| Model | $\text{len}(t_1)$=512 | Heuristic |
|---|---|---|
| DS-R1-Qwen-1.5B | 41.5 | 43.8 |
| DS-R1-Qwen-32B | 70.1 | 70.4 |
| Qwen3-8B | 68.9 | 68.2 |
| SW-OR1-7B | 57.9 | 57.7 |

**Overhead from reward model.** Relative to vanilla sampling, early pruning requires scoring the first step with a reward model, which introduces additional overhead. To ensure a fair comparison, we avoid using extra GPUs for deploying the reward model. Instead, our procedure is as follows: (1) load the reasoning model to generate

Table 4: Time spent by early pruning.

| Model | $N = 64$ | $M = 16$ |
|---|---|---|
| DS-R1-Qwen-1.5B | ×1.00 | ×0.27 |
| Qwen3-8B | ×1.00 | ×0.37 |

candidate first steps and then offload it; (2) load the reward model on the same GPU to evaluate these steps and offload it; and (3) reload the reasoning model to continue generation from the selected first steps. The timing results are reported in Table 4. Notably, early pruning remains efficient both in terms of token usage and runtime, since evaluating the first step with the reward model is inexpensive—comparable to computing embeddings for the short first steps.

## 5 CONCLUSION

In this paper, we empirically demonstrate that the first reasoning step plays a critical role in determining the final outcome of a model's reasoning process. Errors introduced early can significantly degrade overall performance. Motivated by this observation, we propose an efficient sampling strategy that identifies and retains high-quality first reasoning steps, reducing the inference computation up to 70% across three model families. These findings suggest that improving or exploiting the very first step is a promising direction for building more accurate and efficient reasoning LLMs.

## ETHICS STATEMENT

This work builds on publicly available datasets (AIME24, AIME25, HMMT Feb 2025, GPQA, and LiveCodeBench) and openly released models (e.g., DeepSeek, Qwen), following their respective licenses. We report exact model identifiers and dataset subsets to facilitate reproducibility. We acknowledge that improvements in efficiency may reduce computational cost barriers, which could enable misuse (e.g., large-scale automated disinformation). However, our focus is on rigorous evaluation of reasoning processes and improving efficiency for legitimate research purposes. We encourage responsible use of our findings and emphasize that the proposed methods should be applied in research and educational contexts, not for harmful applications.

## REPRODUCIBILITY STATEMENT

Our code builds on widely used open-source libraries for training and evaluation of LLMs, including vLLM and Hugging Face Transformers. All experiments are conducted on datasets available through Hugging Face, and we adopt default prompt formats unless otherwise specified. We report the exact model names and versions used, including both open-source and closed-source models.

For datasets, we evaluate on AIME24, AIME25, HMMT Feb 2025, GPQA, and LiveCodeBench, specifying the subsets employed in our experiments. All datasets are publicly available. We will release our code (and configuration details) upon acceptance of this paper to further support reproducibility.

We encourage the community to design and release additional evaluation datasets that are explicitly intended for research use and distributed under open licenses.

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

## A    LIMITATIONS

Our study has several limitations. First, we restrict our analysis to open-source long-CoT models. This excludes state-of-the-art closed-source systems such as GPT-4 and Claude, where reasoning traces are not accessible and controlled interventions are difficult. Focusing on open-source models ensures transparency and reproducibility, and the insights we uncover are in principle model-agnostic. Second, our experiments focus on mathematical, code generation, and scientific reasoning benchmarks. Although these datasets are limited in scope, they are representative of structured multi-step reasoning and provide a controlled setting for evaluating first-step effects. Importantly, our approach is flexible and could be extended to broader domains such as commonsense reasoning or creative writing with task-specific modifications.

## B    THE USE OF LARGE LANGUAGE MODELS

We used LLMs (e.g., GPT-5) to assist with segmenting reasoning traces into steps as part of our pipeline (see Section 3.1). We also used ChatGPT to generate plotting scripts and to make minor improvements in grammar and phrasing of the manuscript. No LLM was used for research ideation, experiment design, dataset creation, or for producing original scientific contributions. All conceptual and methodological contributions are from the authors.

## C    KEYWORDS FOR REASONING STEP SEGMENTATION

Good reasoning switching keywords should signal a clear switch in reasoning, while should not occur too frequently which often indicate only minor digressions, or too rare to be reliable delimiters. We collect a list of reasoning switching keywords from papers (Muennighoff et al., 2025; Hou et al., 2025). To assess which reasoning switch keywords could serve a similar role, we compute their average frequency per question in model outputs. After cosidering the frequencies and manually check the segment results, we found that *"alternatively"* strikes a balance, appearing often enough to capture step boundaries while still reflecting shifts in reasoning, for DS-R1-Qwen-7B and Qwen3-8B. We uses both *"alternatively"* and *"wait"* for Claude-3.7-Sonnet, GPT-OSS-20B and Magistral-Small.

Table C.1: Average frequency per question of discourse markers in model outputs.

|                  | but   | wait | alternatively | hmm | hold on | let me confirm | however |
|------------------|-------|------|---------------|-----|---------|----------------|---------|
| DS-R1-Qwen-7B    | 82    | 72   | 14            | 7   | 3       | <1             | 10.1    |
| Qwen3-8B         | 91    | 66   | 20            | 6   | 1       | <1             | 10.4    |
| Claude-3.7-Sonnet| 28.5  | 22.6 | 0.2           | 6.1 | 0.7     | 0.0            | 0.9     |
| Magistral-Small  | 47.1  | 54.8 | 3.0           | 0.0 | 0.9     | 0.2            | 1.4     |
| GPT-OSS-20B      | 171.6 | 60.5 | 4.9           | 1.2 | 0.7     | 0.0            | 5.9     |

## D    MODEL IDENTIFIERS

For reproducibility, we provide the exact model identifiers used in our experiments. The models are: DeepSeek-R1-Distill-Qwen-7B (`deepseek-ai/DeepSeek-R1-Distill-Qwen-7B`), Qwen3-8B (`Qwen/Qwen3-8B`), Claude 3.7 (`claude-3-7-sonnet-20250219`, accessed via the Claude API), GPT-OSS-20B (`openai/gpt-oss-20b`), and Magistral-Small (`mistralai/Magistral-Small-2509`).

## E    ADDITIONAL SIMILARITY ANALYSES

We report the cosine similarity between the embedding of the $i$-th reasoning step $t_i$ and the final conclusion $c_T$, interpolated using the maximum number of reasoning steps (Figure E.1).

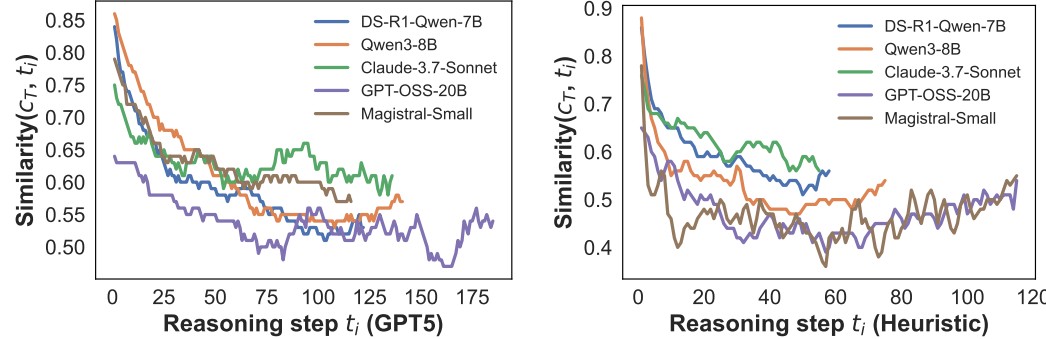

Figure E.1: Cosine similarity between the embeddings of the i-th reasoning step $t_i$ and the final conclusion $c_T$, using the maximum number of reasoning steps for interpolation. The reasoning steps are segmented either by GPT-5 (left) or by heuristic rules (right).

In addition, we report SPLADE similarity (Formal et al., 2021), which provides a sparse lexical–semantic relevance signal by expanding tokens into weighted lexical features. We compute SPLADE similarity using the `naver/splade-cocondenser-ensembledistil` model (Formal et al., 2022), following standard practice. As shown in Figure E.2, the trends closely mirror those observed under cosine similarity. These results serve as complementary checks, confirming that our findings are not tied to dense embedding models and hold across both dense and sparse similarity measures.

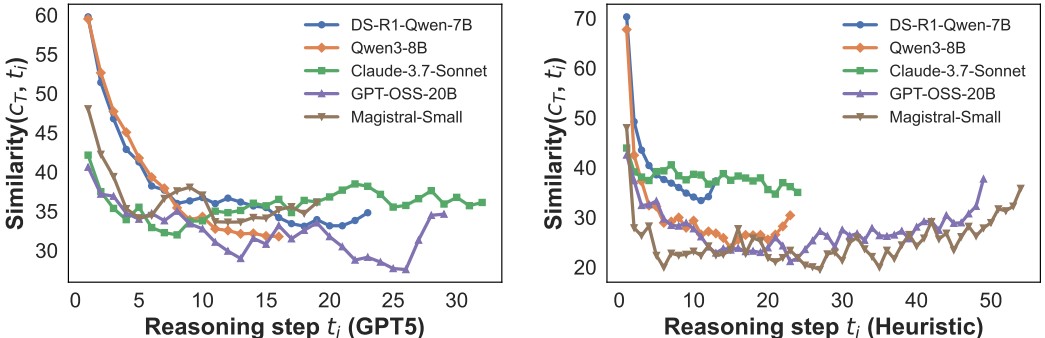

Figure E.2: SPLADE similarity between the representations of the i-th reasoning step $t_i$ and the final conclusion $c_T$, using the maximum number of reasoning steps for interpolation. The reasoning steps are segmented either by GPT-5 (left) or by heuristic rules (right).

## F   CORRELATION BETWEEN FIRST REASONING STEP AND FINAL PREDICTION

In the main text, we reported results on DS-R1-Qwen-7B. Here we extend the analysis to Qwen3-8B (Figure F.1) and further include DeepSeek and Qwen3 models of different sizes (Figure F.2). These additional results verify that the correlation between the first reasoning step and the final prediction persists as model size scales.

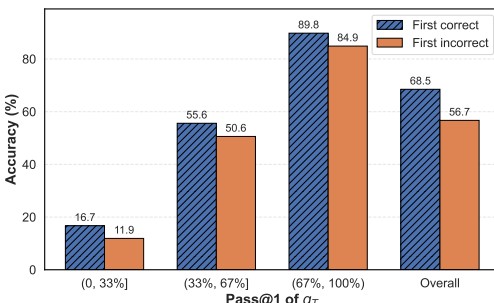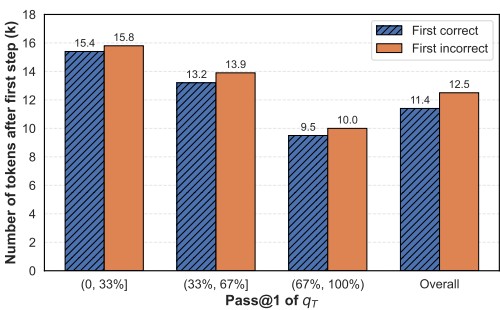

Figure F.1: Accuracy and number of tokens on Qwen3-8B. **Left:** The relationship between the accuracy of the final prediction ($q_T$) and the correctness of the prediction solely based on the first reasoning step ($q_1$) across different difficulty levels. If $q_1$ is incorrect, $q_T$ is more likely incorrect. **Right:** The number of tokens used for the final prediction after the first reasoning step $t_1$, i.e., the number of tokens used for $[t_2, t_3, ..., t_T]$. Although $q_1$ is correct, the model still consumes a large amount of tokens for the following reasoning steps–overthinking.

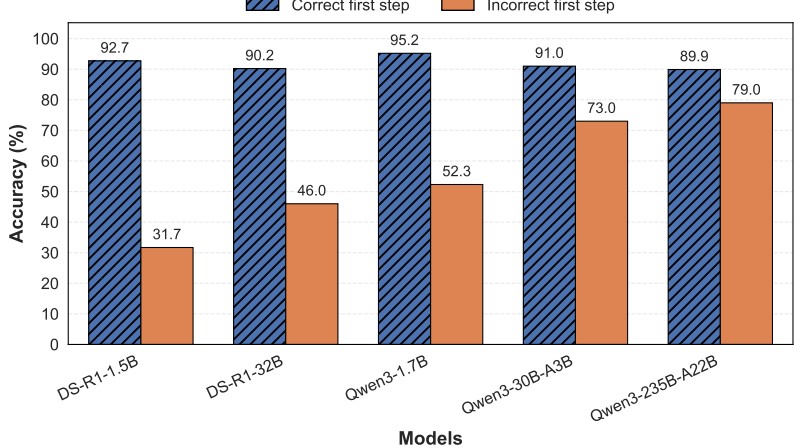

Figure F.2: More results from Qwen and DeepSeek models of different sizes. We report the relationship between the accuracy of the final prediction ($q_T$) and the correctness of the first reasoning step ($q_1$). If $q_1$ is incorrect, $q_T$ is more likely to be incorrect.

## G  PROMPT EXAMPLES

The template for using the DeepSeek-R1 model is provided below. We apply the same structure to other open-source thinking models, with slight adjustments for model-specific input and output formats.

**Prompt example for DeepSeek-R1:**

```
<|begin_of_sentence|>Please reason step by step, and put your
final answer within \boxed{}.

<|User|>[Question]<|Assistant|><think>
```

## H  EXAMPLE OF A INCORRECT FIRST STEP

Figure H.1 illustrates a case where the initial reasoning step is incorrect.

---

**Example of incorrect first step**

**Question:** Four unit squares form a $2 \times 2$ grid. Each of the 12 unit line segments forming the sides of the squares is colored either red or blue in such a way that each unit square has 2 red sides and 2 blue sides. One example is shown below (red is solid, blue is dashed). Find the number of such colorings.
[asy] size(4cm); defaultpen(linewidth(1.2)); draw((0, 0) − (2, 0) − (2, 1)); draw((0, 1) − (1, 1) − (1, 2) − (2,2)); draw((0, 0) − (0, 1), dotted); draw((1, 0) − (1, 1) − (2, 1) − (2, 2), dotted); draw((0, 1) − (0, 2) − (1, 2), dotted); [/asy]
**First reason:** Alright, so I have this problem about coloring the sides of a 2x2 grid of squares. Each square has four sides, and each of these sides is colored either red or blue. The condition is that each individual square must have exactly two red sides and two blue sides. The question is asking for the number of such colorings.
First, let me visualize the grid. It's a 2x2 grid, so there are four squares arranged in a larger square. Each square has four sides, and each side is shared with another square except for the outermost sides . But in the problem, it says there are 12 unit line segments forming the sides of the squares. Wait, let me check that: a 2x2 grid has 3 horizontal lines and 3 vertical lines, each of length 2 units. So, each horizontal line is divided into two segments, right? So, the total number of horizontal segments is 3 rows times 2 segments per row, which is 6, and similarly, the vertical segments are also 3 columns times 2 segments per column, which is another 6. So, 6 + 6 = 12 segments in total. Okay, that makes sense.
Each of these 12 segments can be colored either red or blue. However, the constraint is that each of the four squares must have exactly two red sides and two blue sides. So, even though each segment can be colored independently, we have these local constraints for each square.
I think a good way to approach this problem is by considering each square and the possible colorings for its sides, then seeing how they can fit together without violating the constraints. However, since the coloring of one square's side affects the adjacent squares, this could get complicated.
Maybe I can model this as a graph where each segment is a node, and the squares impose constraints on the nodes (i.e., exactly two red and two blue edges around each square). But I'm not too familiar with graph theory, so maybe that's not the best approach.
**Mistakes in first reasoning step:** Mistake 1 incorrectly assumes that in a $2 \times 2$ grid, each corner square has two shared and two unshared sides, without properly accounting for edge distribution. Mistake 2 introduces unnecessary abstraction by treating segments as "nodes," which obscures the natural structure: the segments themselves are the edges whose colours we choose, and each square simply places a rule on its four incident edges. Working directly with those four constraints is far cleaner than introducing an extra level of abstraction.

Figure H.1: Example of incorrect first step.

# I    EXAMPLES FOR MINOR PERTURBATION

**Original final conclusion**

To solve the problem, we are given triangle $ABC$ with side lengths $AB = 5$, $BC = 9$, and $AC = 10$.
[...]
Step 4: Compute $AP$. Since $P$ lies on line segment $AD$, and $D$ lies outside the circle, the point $P$ lies between $A$ and $D$. Therefore:

$$AP = DA - DP = \frac{325}{22} - \frac{2025}{286}.$$

Find a common denominator:

$$\frac{325}{22} = \frac{4225}{286} \Rightarrow AP = \frac{4225 - 2025}{286} = \frac{2200}{286} = \frac{100}{13}.$$

Final Step: Compute $m + n$.
The reduced form of $AP$ is $\frac{100}{13}$, so:

$$m = 100, \quad n = 13 \Rightarrow m + n = \backslash\text{boxed\{113\}}.$$

Final Answer: \boxed{113}

---

**New correct first reason**

To solve the problem, we are given triangle $ABC$ with side lengths $AB = 5$, $BC = 9$, and $AC = 10$.
[...]
Step 4: Compute $AP$. Since $P$ lies on line segment $AD$, and $D$ lies outside the circle, the point $P$ lies between $A$ and $D$. Therefore:

$$AP = DA - DP = \frac{325}{22} - \frac{2025}{286}.$$

Find a common denominator:

$$\frac{325}{22} = \frac{4225}{286} \Rightarrow AP = \frac{4225 - 2025}{286} = \frac{2200}{286} = \frac{100}{13}.$$

Final Step: Compute $m + n$.
The reduced form of $AP$ is $\frac{100}{13}$, so:

$$m = 100, \quad n = 13 \Rightarrow m + n = 113.$$

---

**New incorrect first reason**

To solve the problem, we are given triangle $ABC$ with side lengths $AB = 5$, $BC = 9$, and $AC = 10$.
[...]
Step 4: Compute $AP$. Since $P$ lies on line segment $AD$, and $D$ lies outside the circle, the point $P$ lies between $A$ and $D$. Therefore:

$$AP = DA - DP = \frac{325}{22} - \frac{2025}{286}.$$

Find a common denominator:

$$\frac{325}{22} = \frac{4225}{286} \Rightarrow AP = \frac{4225 - 2025}{286} = \frac{2200}{286} = \frac{100}{13}.$$

Final Step: Compute $m + n$.
The reduced form of $AP$ is $\frac{100}{13}$, so:

$$m = 100, \quad n = 13 \Rightarrow m + n = 103.$$

Figure I.1: Illustration of a targeted perturbation to create correct and incorrect first reasoning steps, as discussed in §3.3. The modified context is highlighted in orange.

