# OpenReview forum: "Lost at the Beginning of Reasoning"
_ICLR.cc/2026/Conference — ICLR 2026 Conference Withdrawn Submission_

### Official Review · Reviewer_dWmg · 2025-10-25

**Soundness:** 1
**Presentation:** 2
**Contribution:** 2
**Rating:** 2
**Confidence:** 4

**Summary:**

This paper provides evidence that content placed at the beginning of the chain-of-thought strongly influences task outcomes; injecting small errors there can markedly degrade performance. The authors propose a sampling method that scores first-step candidates, expands only the top-M, and then performs self-consistency aggregation—achieving accuracy comparable to vanilla self-consistency with substantially lower token cost.

**Strengths:**

1. Provides empirical evidence on the link between early reasoning and final correctness
2. The idea and presentation are clear: definitions are stated, the pipeline is easy to follow, main figures/tables support the claims, showing the effectiveness of their method.

**Weaknesses:**

1. Ambiguous “beginning of reasoning.” §3.1/§3.2 define the first step via two different segmentations, while §3.3 is actually a prefix intervention (front-loading the conclusion segment). Main experiments also approximate “first step” by a fixed token length. These heterogeneous settings weaken the causal claims.

2. Lack of position analysis for errors. Current evidence does not establish that only beginning errors are disproportionately harmful (as the abstract suggests). Mid- or last-step errors may be similarly damaging. Injecting errors in the middle or at the end of the reasoning could also significantly degrade performance.

3. Closed-source models are mentioned in the abstract but not reported. In addition, the proposed method may not be applicable to proprietary APIs, since decoding/inspecting only the first-step prefix is typically not supported.

4. (minor) Figure 4 axis bug: K is shown as 14, 8, 16, …; please sort numerically (8, 14, 16, 32, 64).

**Questions:**

1. How much of the reasoning does “the beginning” actually cover? Please quantify the proportion (e.g., token share) and give representative examples from the experiments—what information does a 512-token prefix typically contain?

2. In Sec 3.1, “To avoid inflated similarity from problem restatement, we use GPT-5 mini to remove question-overlap text at the beginning of traces.” Please show before/after examples and a few high-similarity cases that don't need cleaning.

3. Do mid- or last-step errors also cause large degradations? Could the authors report results for injecting errors in the middle and near the end for comparison with beginning errors.

4. In Sec 3.3, the authors replace the conclusion number and move that sentence to the very beginning; performance drops sharply. Since this prefix almost constitutes the final answer, how does this establish the importance of the initial stage rather than simply showing strong outcome-anchoring? Please clarify your interpretation.

---

### Official Review · Reviewer_k5Kz · 2025-10-27

**Soundness:** 2
**Presentation:** 3
**Contribution:** 2
**Rating:** 2
**Confidence:** 5

**Summary:**

This paper investigates how individual steps in a chain-of-thought (CoT) contribute to reasoning outcomes in large language models (LLMs). Through empirical analysis, the authors find that the first reasoning step exhibits the highest semantic similarity to the final conclusion and disproportionately determines final accuracy. They show that errors in this step correlate with large accuracy drops (≈ 40 %) across several reasoning-oriented models (DeepSeek-R1, Qwen3, Claude-3.7-Sonnet, etc.). Building on this observation, the paper proposes an early-pruning sampling method that evaluates multiple candidate first steps with a reward model and only continues full CoT generation for the top-scoring ones. This strategy achieves roughly 70 % reduction in inference cost while maintaining accuracy.

**Strengths:**

1. The paper provides a clean and reproducible empirical observation that the first reasoning step strongly correlates with the final answer.
2. The proposed pruning strategy is simple yet effective in improving the original LLM's reasoning performance.
3. Extensive evaluation is conducted on diverse datasets.
4. The writing is mostly clear, figures are illustrative and the experiments are easy to follow.

**Weaknesses:**

1. Limited novelty and conceptual depth.

The main claim (“the first step matters most”) is quite trivial and incremental, especially considering the previous works (e.g. Lost in the Middle, and other efficient CoT and overthinking works) which study the importance of reasoning steps. This claim is only supported by the semantic similarity rather and a causality analysis. Besides, the actual reason why the first step shows more similarity is not explored, because it is possible that both the first step and the final prediction rephrase the original problem statement and then begin to answer. A deeper understanding is needed.

2. The concept of 'reasoning step' in this work seems to be confused. The authors extract steps from the 'thinking token' of reasoning models, which represents the internal thinking of the models. According to the authors' definition, the 'reasoning step' discussed in this work can actually contain a full plan to solve the problem, rather than a single step as commonly defined in CoT. This confusion can hinder readers from getting the idea, especially when the authors do not provide any details of prompts and detailed examples of the experiments.

3. The methodology is also simple and kind of trivial.

There are many step-wise evaluation, selection, and pruning methods, and the detailed methods include using self-reflection, reinforcement learning, and external reward models [1-8]. This work simply proposes to use an external reward model to select the first step, which is a simplified version of selection and pruning for each step. I can hardly see any novel or interesting contribution.

4. The experiments are not convincing. The authors compare with no baselines, which is not sufficient at all, and it is hard to fairly evaluate the experimental contribution of the proposed method.

[1] THINKPRUNE: Pruning Long Chain-of-Thought of LLMs via Reinforcement Learning

[2] s1: Simple test-time scaling

[3] Revisiting the Test-Time Scaling of o1-like Models: Do they Truly Possess Test-Time Scaling Capabilities?

[4] Self-Evaluation Guided Beam Search for Reasoning

[5] DYNAMIC EARLY EXIT IN REASONING MODELS

[6] GenPRM: Scaling Test-Time Compute of Process Reward Models via Generative Reasoning

[7] R-PRM: Reasoning-Driven Process Reward Modeling

[8] Let’s Verify Step by Step

**Questions:**

1. Could the observed “first-step dominance” simply arise because the first step often restates the question or directly starts computing toward the answer? How is this controlled?

2. Have you tested whether later-step corrections (e.g., via reflection or backtracking) can ever override a wrong first step in models explicitly trained for self-correction (e.g., o1-like reasoning models)?

3. How robust is the early-pruning method to reward model bias? If the PRM favors short or confident-sounding first steps, could it systematically prune correct but verbose reasoning?

---

### Official Review · Reviewer_AcHq · 2025-10-30

**Soundness:** 2
**Presentation:** 2
**Contribution:** 2
**Rating:** 2
**Confidence:** 4

**Summary:**

This paper presents an empirical analysis of the pattern of the chain-of-thought reasoning. In particular, it investigates the similarity between each intermediate reasoning step and the final reasoning step. A rough decreasing pattern across the reason step is observed. Finally, a heuristic method that using multiple generation is designed to improve the accuracy of the fist step, which is shown to improve the accuracy of the reasoning.

**Strengths:**

The decreasing pattern in the similarity score between each intermediate reasoning step and the final reasoning step is interesting.

The proposed method to improve the accuracy of the first reasoning step looks reasonable.

**Weaknesses:**

The definition of reasoning step is vague. This paper vaguely states that a reasoning step is a complete logical leap or self-contained unit. Precisely, what is a reasoning step?

The reasoning step segmentation method is questionable. First, the segmentation accuracy is not shown in this paper. This leads to a critical technical flaw. More specifically, if the segmentation accuracy is low, should we trust the observation in this paper? Also, it is possible that the segmented last step does not contain all reasoning details. If that’s the case, what’s the meaning of the similarity score?

The method that eliminates the inflated similarity is also questionable. Removing question-overlap text would lead to certain bias. This treatment lacks convincing justification.

In the methodology, the stopping condition of generating the first reasoning step is missing. The stopping condition is critical.

**Questions:**

See weakness part.

---

### Official Review · Reviewer_vWEJ · 2025-10-30

**Soundness:** 3
**Presentation:** 3
**Contribution:** 3
**Rating:** 6
**Confidence:** 3

**Summary:**

The paper argues that in long chain-of-thought (CoT) reasoning, the very first generated step strongly determines the final outcome. Empirically, when the first step is wrong, final accuracy drops sharply (≈40% on their setups), suggesting limited self-correction in current reasoning LLMs. Building on this, the authors propose early pruning method. That is, sampling many candidate first steps, scoring them with a process reward model (PRM), and only continuing full CoT generation for the top-M candidates. Across multiple model families (DeepSeek-R1, Qwen3, SW-OR1, Magistral, GPT-OSS) and benchmarks (AIME24/25, HMMT Feb 2025, GPQA, LiveCodeBench), this retains accuracy while cutting inference cost by up to 70%. Ablations indicate the specific reward model matters little and that ~512-token first steps are long enough for robust scoring.

**Strengths:**

The paper is well-written, and the idea is interesting. Specifically, it shows that first-step quality heavily correlates with final correctness, and even small perturbations to an otherwise correct first step can yield large accuracy drops. Also, this paper proposes a simple and effective method by early pruning via PRM-scored first steps. Besides, the evaluation covers a lot of models, enhancing the robustness of the evaluation, as well as useful ablation studies. Lastly, the paper uncovers clarity on the overthinking of redundant long traces, even when the first step alone is enough to solve the problem.

**Weaknesses:**

**Limited evaluation datasets for Section 3**: While testing multiple LLMs, the only datasets used for Section 3 are AIME 24 and 25, which only include 60 questions in total. This raises concerns about the robustness of the analysis, given that it is the main finding of this paper. An extended evaluation on other reasoning datasets could make the results more generalizable.

**Pruning potentially contributive segments**: By pruning early, we may discard rare but ultimately correct traces that require later self-correction. It would also be important and promising to test how (i) the later segments alone or (ii) the combination of the first segment with the later segments could correlate to the answer correctness.

**First-step budget and latency**: Generating 64 first steps of ~512 tokens each and scoring them may still be heavy for latency-sensitive settings. Although the paper reports runtime without extra GPUs and shows decent speedups, this could be unexpectedly expensive in harder, practical, or non-English scenarios where the thinking traces could easily exceed 8196 tokens.

**Questions:**

Based on the three main weaknesses, three questions and suggestions are as follows:

(1) Include at least one more reasoning dataset in Section 3.

(2) Verify and evaluate the contributions of the latter segments in the thinking traces. And measure their correlation with the answer correctness.

(3) Discuss the potential solutions for the complex or multilingual questions in practice, where the thinking traces could be much longer. (e.g., one word in Finnish, Turkish, Hungarian could be split into multiple tokens.)

---

### Note · Authors · 2025-12-04

I have read and agree with the venue's withdrawal policy on behalf of myself and my co-authors.